# Comparison of the Material Quality of Al_x_In_1−x_N (x—0–0.50) Films Deposited on Si(100) and Si(111) at Low Temperature by Reactive RF Sputtering

**DOI:** 10.3390/ma15207373

**Published:** 2022-10-21

**Authors:** Michael Sun, Rodrigo Blasco, Julian Nwodo, María de la Mata, Sergio I. Molina, Akhil Ajay, Eva Monroy, Sirona Valdueza-Felip, Fernando B. Naranjo

**Affiliations:** 1Photonics Engineering Group, Electronics Department, University of Alcalá, Madrid-Barcelona Road km 33.6, 28871 Alcalá de Henares, Spain; 2Science, Computations and Technology Department, European University of Madrid, Tajo Street s/n, 28670 Villaviciosa de Odón, Spain; 3Fort Hare Institute of Technology, University of Fort Hare, Alice 5700, South Africa; 4Departamento Ciencia de los Materiales, I.M. y Q.I., IMEYMAT, Universidad de Cádiz, Campus Río San Pedro s/n, Puerto Real, 11510 Cádiz, Spain; 5CEA-Grenoble, INAC/PHELIQS, 17 av. des Martyrs, 38054 Grenoble, France

**Keywords:** AlInN, Si(100), Si(111), radio-frequency sputtering

## Abstract

Al_x_In_1−x_N ternary semiconductors have attracted much interest for application in photovoltaic devices. Here, we compare the material quality of Al_x_In_1−x_N layers deposited on Si with different crystallographic orientations, (100) and (111), via radio-frequency (RF) sputtering. To modulate their Al content, the Al RF power was varied from 0 to 225 W, whereas the In RF power and deposition temperature were fixed at 30 W and 300 °C, respectively. X-ray diffraction measurements reveal a c-axis-oriented wurtzite structure with no phase separation regardless of the Al content (x = 0–0.50), which increases with the Al power supply. The surface morphology of the Al_x_In_1−x_N layers improves with increasing Al content (the root-mean-square roughness decreases from ≈12 to 2.5 nm), and it is similar for samples grown on both Si substrates. The amorphous layer (~2.5 nm thick) found at the interface with the substrates explains the weak influence of their orientation on the properties of the Al_x_In_1−x_N films. Simultaneously grown Al_x_In_1−x_N-on-sapphire samples point to a residual n-type carrier concentration in the 10^20^–10^21^ cm^−3^ range. The optical band gap energy of these layers evolves from 1.75 to 2.56 eV with the increase in the Al. PL measurements of Al_x_In_1−x_N show a blue shift in the peak emission when adding the Al, as expected. We also observe an increase in the FWHM of the main peak and a decrease in the integrated emission with the Al content in room-temperature PL measurements. In general, the material quality of the Al_x_In_1-x_N films on Si is similar for both crystallographic orientations.

## 1. Introduction

Al_x_In_1−x_N ternary semiconductor alloys have attracted huge interest for their application in solar cells, particularly after the revision of the indium nitride (InN) band gap energy in 2001 [1]. The direct band gap (i.e., high absorption coefficient) of Al_x_In_1−x_N, tunable from the near-infrared (0.7 eV for InN [1]) to the ultraviolet (6.2 eV for AlN [2]) range, makes it an excellent candidate for developing photovoltaic devices in combination with silicon. In addition, this material shows high resistance to thermal and mechanical stress and irradiation with high-energy particles [3], which makes it suitable for space applications.

The synthesis of high-quality single-phase AlxIn_1−x_N layers is challenging due to the large difference in properties such as bonding energy, lattice constants, or growth temperature between the binary constituents, InN and AlN. The growth of AlxIn_1−x_N layers has been reported by various techniques, including metal–organic chemical vapor deposition (MOCVD) [4,5,6,7], molecular beam epitaxy (MBE) [8,9,10,11,12], elemental stacks annealing (ESA) [13,14], and reactive sputtering. Within this last technique, we can distinguish two approaches, one that uses a mixture of argon and nitrogen for the deposition [15,16,17,18,19,20,21,22,23,24] and another that uses only nitrogen [25,26,27,28,29,30,31], the latter being our case. Reactive sputtering also allows the deposition on large substrates and employs lower temperatures than MOCVD or MBE. However, the low-temperature deposition comes at the price of higher defect density. The presence of impurities such as hydrogen [32] and defects such as nitrogen vacancies [33] induces unintentional doping with a residual carrier concentration as high as 10^21^ cm^−3^, which causes a blue shift in the optical band gap due to the Burstein–Moss effect [34].

AlxIn_1−x_N can be synthesized on different substrates, such as Si(111) [22,30,31,35,36,37], Si(100) [19,23,27,38], sapphire [17,22,26,30,35,37,38], glass [17,22,30,37], and GaAs [22]. However, the properties of the Al_x_In_1−x_N films strongly depend on the nature of the substrate. It is particularly interesting to study the deposition of silicon due to its potential for hybrid III-nitride/Si solar cells. Wurtzite III-nitrides are usually grown on silicon (111) due to the hexagonal symmetry of this crystallographic plane. However, today, silicon-based nanotechnology uses silicon (100) because of its lower amount of dangling bonds, which generate undesired recombination centers [34].

There are several studies about the growth of Al_x_In_1-x_N films on either Si(111) or Si(100) and its comparison with Al_x_In_1-x_N on sapphire substrates. Bashir et al. [35] deposited InN on Si(111) by RF sputtering and obtained large crystallite size, low microstrain, and low dislocation density. Afzal et al. [22] grew Al_x_In_1−x_N films on Si(111) at 300 °C using a magnetron cosputtering system and obtained polycrystalline films with the preferred orientation along the (101) direction, with higher crystallite size and lower surface roughness compared with other substrates such as GaAs and glass. However, the comparison of Al_x_In_1−x_N layers simultaneously grown on both Si(111) and Si(100) substrates by reactive RF sputtering has never been reported so far. 

This work presents the study of the properties of Al_x_In_1−x_N layers with Al content ranging from 0% to 50% simultaneously deposited on silicon (100) and (111) via reactive RF sputtering at a relatively low substrate temperature, 300 °C. The layer characteristics in terms of structural, morphological, electrical, and optical properties are studied and compared considering both substrate orientations. Finally, we demonstrate the possibility of using silicon (100) as a feasible substrate for developing Al_x_In_1−x_N layers for device applications by taking advantage of its compatibility with today’s silicon-based nanotechnology.

## 2. Materials and Methods

Al_x_In_1-x_N layers were simultaneously deposited in a reactive RF magnetron sputtering system (AJA International, ATC ORION-3-HV, Scituate, MA, USA) on three substrates: *p*-doped 375 µm thick Si(100), *p*-doped 500 µm thick Si(111) (both with a resistivity of 1–10 Ωcm), and on 500 µm thick (0001)-oriented sapphire. This system was equipped with a 2 inch confocal magnetron cathodes of pure In (4N5) and pure Al (5N). The base pressure of the system was in the order of 10^−7^ mbar. The substrate-target distance was fixed at 10.5 cm, and the temperature during the deposition was monitored with a thermocouple placed in direct contact with the substrate holder. The substrates were chemically cleaned in organic solvents before being loaded in the chamber, where they were outgassed for 30 min at 550 °C and then cooled down to the growth temperature. Prior to the deposition, the surface of the targets and the substrates were cleaned using a soft plasma etching with Ar (2 sccm and 20 W), causing no damage to the surface. Al_x_In_1−x_N layers were deposited in a pure N_2_ atmosphere with a nitrogen flow of 14 sccm and a pressure of 0.47 Pa. The RF power applied to the Al target, P_Al_, was set to 0, 100, 150, 175, and 225 W (samples M1–M5, respectively), while the RF power applied to the In target and the temperature were fixed to 30 W and 300 °C, respectively. A sputtering time of 3 h was used for the InN sample, 5 h for the sample with P_Al_ = 100 W, and 4 h for the rest. The thickness and deposition rate of the samples are summarized in Table 1.

The alloy mole fraction, crystalline orientation, and mosaicity of the films were evaluated by high-resolution X-ray diffraction (HRXRD) measurements using a PANalytical X’Pert Pro MRD system (Malvern, UK). In addition, the thicknesses of the layers were obtained from field-emission scanning electron microscopy (FESEM) images. Atomic force microscopy (AFM) was employed to study the surface morphology and estimate the root-mean-square (rms) surface roughness using a Bruker multimode Nanoscope IIIA microscope in tapping mode (Billerica, MA, USA). Additionally, transmission electron microscopy (TEM) provided a deeper understanding of the structural properties of the interface between the deposited material and the substrate. The electrical properties of the films were analyzed using room-temperature Hall-effect measurements in a conventional Van der Paw geometry. 

Finally, photoluminescence measurements were carried out at room temperature by exciting the samples with ~20 mW of a continuous-wave laser diode emitting at λ = 405 nm focused on a 1 mm diameter spot. The emission was collected with a 193 mm focal-length Andor spectrograph equipped with a UV-extended silicon-based charge-coupled-device (CCD) camera operating at −65 °C between 200 and 1100 nm. 

## 3. Results and Discussions

### 3.1. Structural Characterization

To study the structural quality of the layers, HRXRD 2θ/ω scans were carried out on the Al_x_In_1−x_N layers grown on Si(100) and Si(111), with the results shown in Figure 1a,b, respectively. All layers presented a wurtzite crystalline structure highly oriented along the c-axis, and no other crystallographic phases were detected. The increase in P_Al_ shifted the (0002) and (0004) reflection peaks assigned to Al_x_In_1−x_N toward higher diffraction angles, which confirmed the reduction in the c lattice parameter. The Al mole fraction of the alloy was estimated by applying Vegard’s Law [39] to the AlN-InN system, using the c lattice parameter obtained from HRXRD and assuming fully relaxed layers. The calculated Al mole fraction, x, scales linearly with P_Al_ between x = 0 and x = 0.49 or 0.48, for Si(100) and Si(111) substrates, respectively, as summarized in Table 1.

The FWHM of the ω-scan (rocking curve) of the (0002) Al_x_In_1−x_N diffraction peak provides information about the mosaicity of the material. In this study, Al_x_In_1−x_N layers grown on both silicon substrates showed similar values, in the 3–6° range, without a clear trend (Table 1). This indicated that the mosaicity is independent of the crystal orientation of the silicon substrate.

### 3.2. Morphological Characterization

In order to investigate the morphology of the layers, they were studied by FESEM and AFM techniques. Figure 2a–c show the FESEM images of samples grown on Si(100) and Si(111). The morphology of the layers evolved from nanocolumnar for pure InN (sample M1) toward grain-like compact when increasing the Al content (samples M3 and M5) for both substrate orientations. Such a trend was already observed in similar Al_x_In_1−x_N samples deposited on Si(111) by RF sputtering (40 W In, 300 °C) with similar Al compositions [31]. The observed phenomena could be attributed to changes in the surface diffusion of adatoms due to the increased kinetic energy of the incoming Al species, which can determine the layer morphology for both substrate orientations. 

The observed morphological transition was accompanied by a modification of the sample surface roughness. The rms roughness was measured by AFM images scanned in a 2 × 2 μm area (Figure 3). The results showed a surface roughness evolution from 11.5 (Al content x = 0) to 2.5 nm (x ≈ 0.36) for Si(100) and from 13.0 (Al content x = 0) to 2.5 nm (x ≈ 0.36) for Si(111), as summarized in Table 1, and in agreement with previously published results [31]. The roughness remained almost constant for samples with an Al content in the range within x ≈ 0.36–0.42 (see Table 1), and it finally dropped up to ≈2.5 nm for an Al content of ≈50%. This surface roughness reduction was attributed to an increase in the adatom energy and mobility when increasing P_Al_, in agreement with results obtained in similar Al_x_In_1-x_N-on-Si(100) samples deposited at a higher temperature (550 °C) [27], where the surface roughness was 2.0 and 1.5 nm for x ≈ 0.35 and x ≈ 0.56, respectively. 

The interface between the Al_x_In_1−x_N and the silicon substrate was studied by transmission electron microscopy (TEM) measurements. Figure 4 shows the cross-sectional TEM images of an Al_x_In_1−x_N (x ≈ 0.36) layer deposited on Si(100) and Si(111), evidencing the epitaxial growth along the c-axis for the two silicon orientations. In both cases, the images reveal the formation of an amorphous layer of ~2.5 nm at the layer/substrate interface (see the inset of both figures), which may have weakened the interactions between phases and reduced the influence of the silicon orientation on the quality of the nitride layer deposited on top. The similar structural quality obtained growing on both substrates was also confirmed by the comparable grain size estimated from STEM images (Figure 5a,b). Thus, the structural quality is conserved even when grown on a cubic substrate, although a clearer boundary between the amorphous interfacial layer and the nitride one was observed in this case. 

### 3.3. Electrical Characterization

The electrical properties of the Al_x_In_1-x_N layers could only be addressed for samples deposited on sapphire substrates, because the silicon conduction masked the layered signal whenever a silicon substrate was used. The values of resistivity, carrier concentration, and mobility were obtained for simultaneously grown layers with an Al content up to 0.32. Samples with higher Al content showed a resistivity above 10 mΩ·cm, making the Hall effect measurement unreliable.

The layer resistivity increased from 0.38 mΩ·cm for InN to 8 mΩ·cm for Al_0.32_In_0.68_N, while the carrier concentration decreased from 1.73 × 10^21^ cm^−3^ for InN to 2.48 × 10^20^ cm^−3^ for Al_0.32_In_0.68_N. On the other hand, the values of mobility showed no clear trend, starting with a value of 9.5 cm^2^/V.s for InN and decreasing to 3.2 cm^2^/V.s for Al_0.32_In_0.68_N with a peak of 11.5 cm^2^/V.s for Al_0.14_In_0.86_N. The values of resistivity and mobility obtained for the Al_0.32_In_0.68_N sample are similar to those reported by Liu et al. [38] (1.2 mΩ·cm and 11.4 cm^2^/V·s, respectively, for a ~90 nm Al_0.28_In_0.72_N layer deposited by RF sputtering at 600 °C). The high carrier concentration of the layers is related to the unintentional doping from impurities such as hydrogen or oxygen during growth [32], and it was also observed by Nuñez-Cascajero et al. [26], where similar Al_x_In_1−x_N on sapphire with homogeneous distribution of oxygen were obtained. 

### 3.4. Optical Characterization

The apparent optical band gap energy of the samples deposited on sapphire was estimated through room-temperature optical transmittance measurements following the procedure described in Ref. [26] (See Table 2 for all optical data). Figure 6 shows the squared absorption used for this estimation, obtained from the transmittance spectra depicted in the inset of the figure for each sample.

As expected, the apparent optical band gap energy blue shifted with the Al mole fraction as following: Eg_Abs_ ~ 1.70 eV for InN (M1), 1.80 eV (M2), 2.10 eV (M3), 2.30 eV (M4), and 2.60 eV for Al_0.43_In_0.57_N (M5). This blue shift in the optical band gap of the InN, compared with the theoretical of 0.7 eV, was attributed to the high residual carrier concentration of the layer.

Figure 7 shows the low-(11 K) and room-temperature (300K) PL emission of samples M1 (InN) and M2 (Al_x_In_1−x_N, x—0.12, 0.16), grown on Si(100) and Si(111). No PL emission was observed for Al_x_In_1−x_N layers with higher Al content than 16%, independent of the crystal orientation of the substrate. The results obtained from the analysis of the PL measurements in terms of the main peak emission energy, FWHM, and integrated intensity are summarized in Table 3. 

The dominant room-temperature emission energy centered at ≈1.60 and ≈1.80 eV for the M1 (InN) and M2 (Al_x_In_1−x_N, x—0.12, 0.16) samples deposited on both silicon substrates, respectively. The position of the emission energy practically stayed the same, while the intensity decreased when increasing the temperature from 11 to 300 K, as expected. However, the presence of an emission at room temperature was a clear indication of the good crystalline quality of the samples. The FWHM of the PL emission of the samples was similar for both types of substrates, being slightly higher for sample M2, probably due to the alloy disorder present in the Al_x_In_1−x_N layer. 

Then, assuming that the band gap energy was similar for the samples grown on Si and sapphire, we could extract an approximate value for the Stokes shift as the difference between the band gap energy obtained from transmission measurements (Eg_Abs_) and the PL emission energy (E_PL_) at 300 K. The obtained Stokes shift was around ~130 and ~60 meV for InN (M1) and Al_x_In_1−x_N (M2), respectively. These values pointed to a reduced band tail for the Al_x_In_1−x_N samples compared with the InN ones, which could be related to the change in layer morphology (and probably the surrounding of the involved emission centers) when introducing aluminum into the InN binary. 

Lastly, comparing each sample on both substrates, they showed a very similar emission shape and integrated intensity, even though the Al_x_In_1-x_N ones had double the layer thickness compared with their InN counterparts. This result pointed to an enhancement of the nonradiative recombination channels due to Al incorporation, which could increase the lattice disorder and defects.

## 4. Conclusions

Al_x_In_1−x_N films with low-to-mid Al content (*x*—0–0.50) were deposited via RF sputtering on different substrates, i.e., Si(100) and Si(111), for their comparison. The increase in the Al mole fraction improved the structural and morphological quality of the layers, achieving a minimum FWHM of the (0002) Al_x_In_1−x_N rocking curve of ~2.8° and a minimum rms surface roughness of ~2.5 nm for samples grown on both Si substrates with *x*—0.49. FESEM images showed a morphological transition from nanocolumnar toward a grain-like compact morphology when aluminum was introduced. Cross-sectional TEM images revealed a ~2.5 nm thick amorphous layer in the interface between the nitride material and the substrate, which could be responsible for the weak coupling between the active layer and the substrate. This finding allows the development of Al_x_In_1−x_N with similar material quality on both silicon substrate orientations. 

Hall-effect measurements revealed a carrier concentration above 10^20^ cm^−3^ for the Al_x_In_1-x_N layers with x < 0.32, probably induced by the unintentional doping of the material during deposition. Additionally, the Al_x_In_1-x_N layers (x ≤ 0.16) deposited on both Si substrate orientations exhibited similar PL emission in terms of shape, energy, FWHM, and integrated intensity at room temperature, showing a reduction in the PL emission efficiency when introducing the Al compared with the one obtained for InN layers. 

In this work, we demonstrated the ability to produce high-quality Al_x_In_1−x_N layers on Si with low-to-mid Al content via RF sputtering regardless of the chosen substrate orientation.

## Figures and Tables

**Figure 1 materials-15-07373-f001:**
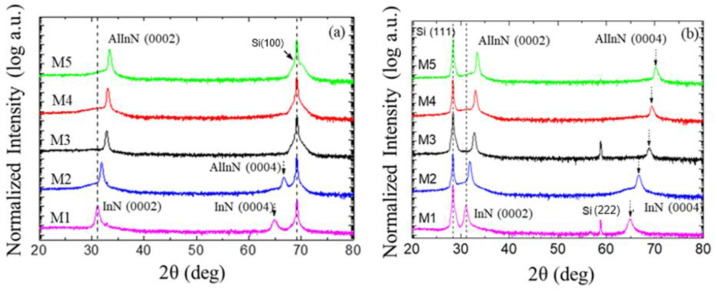
The 2θ/ω scans of the Al_x_In_1−x_N layers deposited on (**a**) Si(100) and (**b**) Si(111) for different P_Al_. The only reflections assigned to Al_x_In_1−x_N were (0002) and (0004). The rest of the reflections were assigned to the substrates.

**Figure 2 materials-15-07373-f002:**
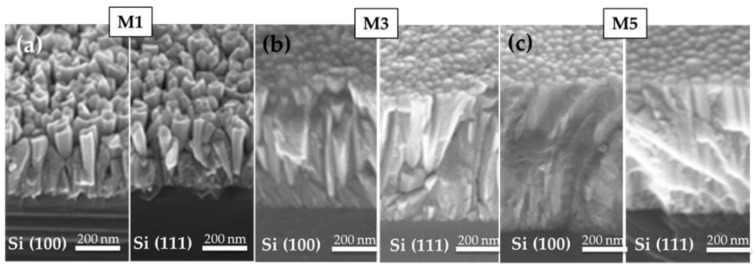
FESEM images of Al_x_In_1−x_N samples (**a**) M1, (**b**) M3, and (**c**) M5 on Si(100) (**left**) and Si(111) (**right**).

**Figure 3 materials-15-07373-f003:**
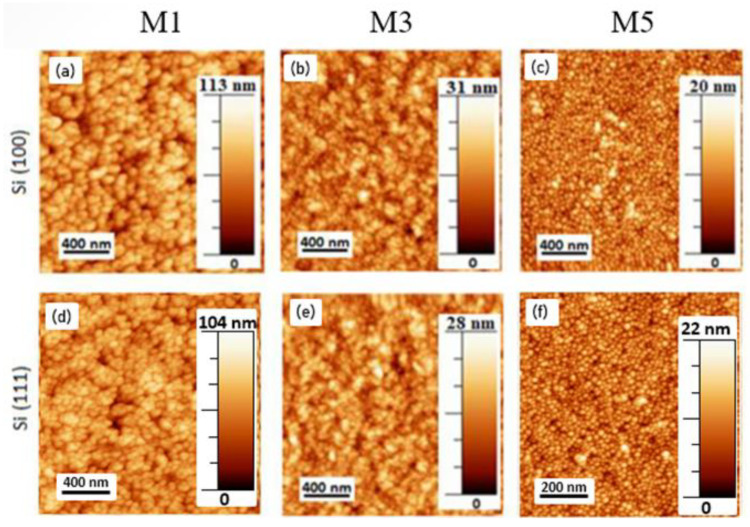
AFM images with a scanning area of 2 × 2 μm of InN and Al_x_In_1−x_N samples with P_Al_ = 0, 150, and 225 W grown on Si(100) (**a**–**c**) and Si(111) (**d**–**f**).

**Figure 4 materials-15-07373-f004:**
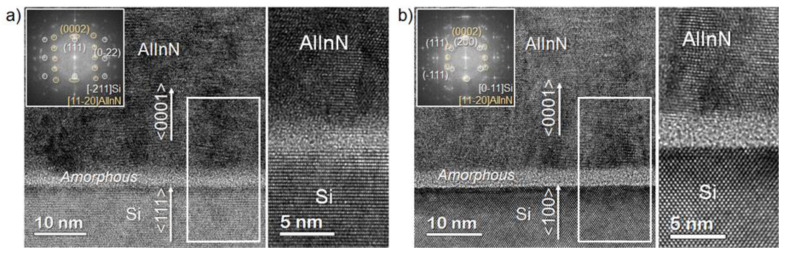
HRTEM images of Al_0.36_In_0.64_N samples grown on (**a**) Si(100) and (**b**) Si(111), along with magnified details of the interphase (right side). Insets show the epitaxial relationship between the layer and substrate.

**Figure 5 materials-15-07373-f005:**
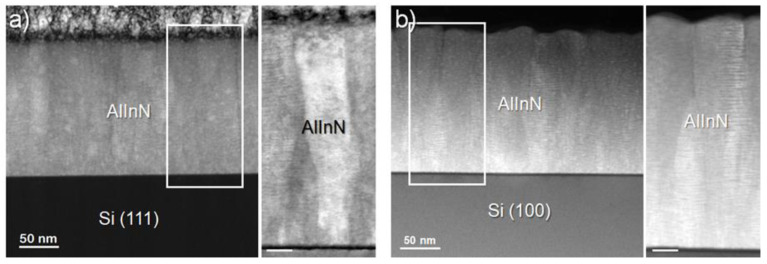
HRTEM images of Al_0.36_In_0.64_N samples grown on (**a**) Si(100) and (**b**) Si(111) show a similar grain size on both substrates. Scale bar at the magnified details is 20 nm.

**Figure 6 materials-15-07373-f006:**
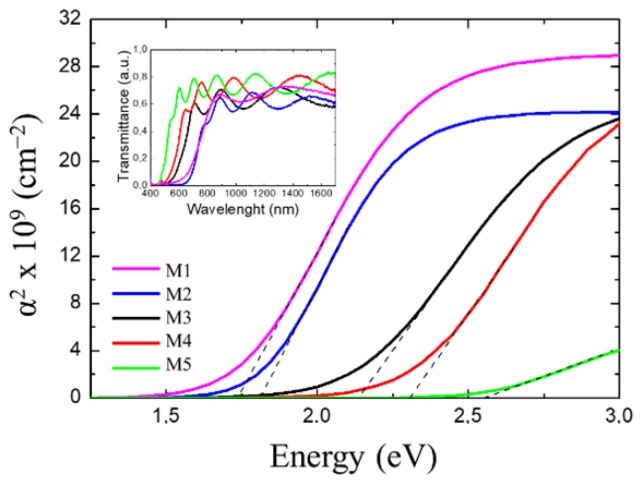
Squared absorption coefficient α^2^ as a function of the energy extracted from the sigmoidal approximation of the Al_x_In_1−x_N layers grown on sapphire. Dashed lines are the linear fits used to estimate the apparent optical band gap energy of the samples E_g_^Abs^. Inset: transmittance spectra vs. wavelength of the same samples M1–M5.

**Figure 7 materials-15-07373-f007:**
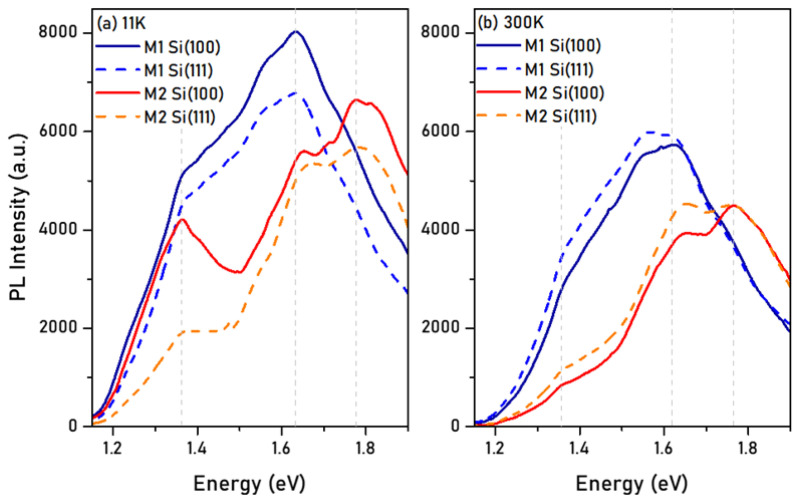
(**a**) Low-temperature (11K) and (**b**) room-temperature (300 K) PL emission of Al_x_In_1−x_N layers deposited on Si(100) and Si(111). For x > 0.16, no PL emission was observed.

**Table 1 materials-15-07373-t001:** Summary of the deposition parameters and the structural and morphological analysis of Al_x_In_1−x_N on Si(100) and Si(111): c-axis parameter and Al mole fraction x extracted from HRXRD, layer thickness estimated from FESEM, and rms surface roughness measured by AFM.

Sample	Substrate	P_Al_ (W)	c (Å)	Al Mole Fraction *x*	FWHM Rocking Curve (°)	Thickness ^1^ (nm)	Deposition Rate ^2^ (nm/h)	Rms Surface Roughness ^3^ (nm)
M1	Si(100)	0	5.73	0	4.6	390	130	11.5
M2		100	5.61	0.12	2.4	790	160	9.5
M3		150	5.45	0.35	6.2	650	160	3.5
M4		175	5.42	0.40	3.2	620	155	3.5
M5		225	5.36	0.48	2.8	910	230	2.5
M1	Si(111)	0	5.73	0	4.7	380	125	13.0
M2		100	5.59	0.16	2.9	780	160	8.0
M3		150	5.44	0.36	6.1	640	160	3.5
M4		175	5.40	0.42	3.1	630	160	3.5
M5		225	5.35	0.49	2.8	585	150	2.5

^1^ Standard error of ±30 nm. ^2^ Standard error of ±15 nm/h. ^3^ Standard error of ±0.3 nm.

**Table 2 materials-15-07373-t002:** Summary of the optical transmittance characterization at room temperature: apparent optical band gap energy (E_g_^Abs^), absorption band edge broadening (ΔE), and linear absorption well above the band gap ($\alpha_{0}$) of the samples under study.

Sample	Al Mole Fraction *x*	$\alpha_{0}\;\left(\times 10^{4}\;cm^{-2}\right)$	$E_{g}^{Abs}\left(eV\right){\;}^{1}$	$\Delta E\left(meV\right){\;}^{2}$
M1	0	17.2	1.70	160
M2	0.12	20.3	1.80	120
M3	0.35	18.4	2.10	210
M4	0.40	18.3	2.30	210
M5	0.48	10.0	2.60	180

^1^ Standard error of ±0.03 eV. ^2^ Standard error of ±10 meV.

**Table 3 materials-15-07373-t003:** Summary of the analysis of the PL measurements at 11 K and 300 K of InN (M1) and Al_x_In_1−x_N (M2) on Si(100) and Si(111).

Sample	Temperature (K)	Substrate	Main Peak Emission Energy ^1^ (eV)	FWHM ^2^ (meV)	Integrated Intensity ^3^ (a.u.)
M1	11	Si(100)	1.60	560	4500
		Si(111)	1.60	515	3600
	300	Si(100)	1.60	460	2750
		Si(111)	1.60	465	2920
M2	11	Si(100)	1.80	565	3750
		Si(111)	1.80	480	3100
	300	Si(100)	1.80	500	2250
		Si(111)	1.75	490	2400

^1^ Standard error of ±0.05 eV. ^2^ Standard error of ±5 meV. ^3^ Standard error of ±10.

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
