# Peer review of "Comparison of the Material Quality of AlxIn1−xN (x—0–0.50) Films Deposited on Si(100) and Si(111) at Low Temperature by Reactive RF Sputtering"

_materials, 2022, doi:10.3390/ma15207373_

Round 1

Reviewer 1 Report

Report on Comparison of the material quality of AlxIn1-xN (x ~ 0-0.50) 2 films deposited on Si(100) and Si(111) at low temperature by re-3 active RF sputtering by Michael Sun et al.

This submission presents work dealing with the fabrication of thin films of AlxIn1-xN radio-frequency sputtering on different substrates, characterization prepared films by standard methods and study of optic and electronic properties.

The paper could be of interest for potential readers. On the other hand, below I suggest some points, which should be addressed by the authors:

1.       The samples were apparently measured by FESEM. There are some results regarding the thickness of these materials in Table 1, but compositions of prepared films were not provided.

2.       Thickness of prepared samples are highly different. These differences can affect quality of surface. For this reason, would be appropriate change the deposition time (when deposition rates are known)

3.       Error bars are missing completely.

4.       Other relevant works dealing with AlInN thin films need to be mentioned, such as Afzal et al. Modern Physics Letters B Vol. 29, No. 28, 1550169 (2015) or Materials Research Express 1 (2014) 026403.

5.       How many AFM measurements were performed on each sample?

6.       Inaccurately names of chapters- preferable- Chapter 3- Results and discussion, Chapter 4- Conclusions.

In conclusion, I recommend publication of this submission in Materials after minor changes.

Reviewer 2 Report

In the paper authored by Michael Sun et al., authors investigated dependence of Al content on properties of sputtered AlInN films. The paper is well organized and easy to follow. However I recommend a few changes before its publication:

- please use subscript for AlxIn1-xN and version with numbers

- lines 208-217 are repletion of text from lines 145-154

- did authors investigated the samples with help of some depth dependent measurements? I mean SIMS or ECV for instance. It would be very interesting to examine if properties (composition, carrier concentration) change with sample depth profile

- please discuss in more detail a very high carrier concentration in obtained films. Please refer to similar papers where such high intrinsic concentration was obtained.

- Table 2, what is a definition of absorption band edge broadening?

- Figure 7 and Table 3. Why the position of PL maximum does not change with temperature (300K vs 11K)? In my opinion AlInN have to follow the Varshini equation. See for instance the paper Temperature dependence of the optical properties of AlInN, Journal of Applied Physics 106, 013515 (2009); https://doi.org/10.1063/1.3160299

Round 2

Reviewer 2 Report

I appreciate authors correction and answers on my remarks.

In the revised version of the paper are Figures are missing.

Author Response

Thank you for the comment. Regarding the missing figures, I can't find it. The original article had 7 figures and the final reviewed manuscript has them all.